# An Exploratory Study of Mobile-Based Scenarios for Foreign Language Teaching in Early Childhood

Markos Konstantakis *[ID], Aggeliki Lykiardopoulou, Electra Lykiardopoulou, Georgia Tasiouli and Georgios Heliades

Department of Digital Media and Communication, Ionian University, 28100 Kefalonia, Greece; g20lyki@ionio.gr (A.L.); lykiardopoulou@ionio.gr (E.L.); d17tasi@ionio.gr (G.T.); heliades@ionio.gr (G.H.)
* Correspondence: konstadakis@ionio.gr or mkonstadakis@aegean.gr

**Abstract:** In today's world, the ability to communicate in a foreign language is more highly prized than ever by prospective employers, which results in more options and possibilities for students, both academically and professionally. As a result of this tendency and the need for new communication methods, language instructors are driven to include cutting-edge language teaching approaches, resources, and materials in their classroom instruction, such as using ICT, or information, communication and ubiquitous technologies. In this paper, we introduce learning scenarios based on two mobile learning apps that facilitate language learning through interesting, interactive settings in a more personalized way based on children's age. The writers' emphasis will be on demonstrating interactive activities devised in their classrooms and on providing examples of student work in two languages, English and Spanish. Through this paper, we examine a range of educational tools and determine that Mondly Kids and Language Drops-Kahoot are the best acceptable teaching materials. On the basis of this assumption, we created three distinct groups of students, and the outcomes from the assessment technique show that mobile language learning enhances children's experiences and increases their willingness to learn a new language. Additionally, students can use mobile applications to improve their speaking abilities and critical thinking skills throughout a language learning session.

**Keywords:** digital learning design; courseware design; language learning; educational tools; interactive learning environments; mobile learning

## 1. Introduction

Personalized learning via mobile devices is a recent trend gaining traction on a global scale, offering new avenues for enhancing and promoting language learning. Mobile learning has grown in popularity due to the increasing usage of smartphones and other personal devices. Many teachers have accommodated mobile technology as a means of facilitating individualized learning. Nevertheless, certain concerns and questions have been raised about access, security, affordability, teacher and learner proficiency in using digital tools [1], and prospective hazards posed by more intelligent technologies [2].

One thing is certain: mobile devices are handy instruments frequently utilized for informal education. The widespread use of phones, smartphones, tablets, and wearables for everyday activities demonstrates both the broad appeal of personalized learning and the potential to advance the concept of personalization in education at all levels. This article intends to assist teachers in considering how personalized learning might contribute to the improvement of language teaching and learning. A high-level overview of digital educational tools is offered, along with explaining the importance of personalization in mobile learning.

In classrooms worldwide, researchers have collaborated with teachers on a plethora of mobile learning projects aimed at assisting groups of students in mastering difficult concepts—which, for many people, includes language learning—and enabling them to reap

the full benefits of educational opportunities. Numerous projects and initiatives focus on the learner and aim to facilitate a more personalized learning experience. Personalization has been a central theme in research and practice in mobile learning, even if it has not always been labelled as such [3,4].

According to recent research [5–11], mobile learning can be both intrinsically and extrinsically motivating and promote cooperation and collaborative work. Our Generation Z students are constantly connected via social media and carry mobile devices wherever they go. As a result, teachers must engage students in their connected, collaborative world to increase the meaning and relevance of their learning. This active involvement in language learning and teaching entails that learners take greater responsibility for their learning and that teachers facilitate and support this learning.

In this article, we thoroughly evaluate different mobile software tools based on their educational features. The consequent comparison among these tools leads us to choose two of them, Mondly AR$^©$ (v.1.1, ATi Studios, Romania) and Drops Language-Kahoot$^©$ (v.36.20, Plans Labs, Korea), which are used as showcase tools for our teaching practices. We then use the chosen tools to elaborate on how they could be used in a learning setting, exhibit how to create engaging activities with these apps, and demonstrate how to use them in an English and Spanish language classroom. Additionally, we will share examples of student work and offer suggestions, tips, and recommendations based on three classroom experiences with these tools.

The contribution per section and the structure of this paper are as follows: Section 2 is devoted to a thorough examination of digital technologies in language learning. The methodological approach is presented in Section 3, which includes a concise and precise comparison of educational tools and selecting the final tools for implementation. Section 4 shows how to evaluate the proposed digital tools in two languages (English and Spanish) using a case study in three schools with children of various ages. Finally, Section 5 concludes the paper.

## 2. Digital Technologies in Language Learning—A Systematic Review

This section presents the background knowledge necessary for understanding the concept of digital and mobile technologies in language learning through a personalized approach.

### 2.1. Digital Storytelling

In the bibliography, the definition of digital storytelling has prevailed and is referred to as "the combination of traditional oral narration with 21st-century multimedia and modern communication tools" [12]. It has emerged in recent years as a powerful teaching and learning tool that attracts both teachers and their students. Its dynamics are due to the fact that stories can be produced by any person, anywhere, on any subject and shared online around the world [13]. The dynamics of images and photographs, in combination with music and movement, make digital storytelling an important digital tool for students of all levels and all subjects [14]. It can be used as a learning strategy in all educational contexts and combined with various learning strategies such as play [15,16].

Digital storytelling makes the learning process more interesting and motivates students through individual or group activities. Through storytelling, teachers transfer knowledge to students pleasantly, making the educational process entertaining and creative, and freeing it from all kinds of passivity and conventionality [17]. Teachers make use of children's preferences by laying the groundwork for developing various forms of literacy. For their part, students develop communication skills, learn to construct narratives, creative practice writing and at the same time increase their digital knowledge and skills. They become familiar with multimedia and interactive storytelling and small group work [13]. They develop their personal and narrative discourse while at the same time enhancing their writing skills. Students encounter digital media and experientially create their own stories, acquiring a positive attitude towards technology. They develop literacy skills through

designing, writing, and presenting their digital stories. Their critical thinking and skills such as collaboration, communication, creativity and innovation are cultivated. Digital narratives can be published on websites, blogs, repositories, communities and shared and disseminated of good practice [18,19].

In addition to the many advantages, the literature also mentions the disadvantages of using digital stories in education. In particular, it is stated that digital storytelling can disorient teaching, with the result that it is treated by children to a large extent as entertainment and not as education. There is a risk that the subject of the stories will not be interesting and, therefore, will not be able to pique the children's interest [20]. Lack of technological equipment, lack of teacher training, technical difficulties are other obstacles that prevent teachers and students from getting involved in creating multimodal stories and making use of technology. Finally, when teaching through digital storytelling, it is not possible to take into account and identify all the learning difficulties of students [21].

The digital tools available for the implementation of digital storytelling are of great variety, with the result that there is a possibility of choice by each teacher depending on the capabilities, age, needs, inclinations and interests of his students [22]. Some of the most popular, based on previous research [23,24], are:

- **Story jumper** (https://www.storyjumper.com/) (accessed on 30 March 2022): Story jumper is an open-source software, also called "writing package", suitable for teachers. It enables the gradual construction of multimedia teaching material. At the same time, it is a challenge for the teacher of every level. It can be described as a modern, creative tool that helps in writing modern e-books, which have many pages, texts and superscripts, images and designs, music and videos. Story jumper is one of the online digital storytelling tools used by millions of educators and students of all ages around the world. It is a collaborative storytelling site that incorporates three ideas: creating, reading and sharing. High-quality visual software, user-friendly interface and attractive student creation environment. The ease of use and the ability to share/publish the story are the two key features that have made it known to the educational community that shows a preference for it. The user can create the stories from scratch, starting from the cover of the book where he will choose the title, then from the graphics library, he will use characters, images, different backgrounds and designs. This software enhances collaborative learning as it allows teachers and students to receive feedback. The application is free, requires registration (sign up) and supports the Greek language. It has an easy-to-use and simple interface, so it is ideal for young children with obvious pedagogical use. However, it can also be used by older students to create digital narratives on social, environmental, cultural and other issues such as bullying.
- **Storybird** (https://storybird.com/) (accessed on 30 March 2022): It is a free online learning tool for creating digitally illustrated stories (creative writing) that can be published (embedded) on other websites. The application allows anyone to create visual stories in a very short time, by registering on the site before use. It provides a wide variety of images, which the user can combine and enrich with text in order to create his own illustrated online story. It can be used by students of all ages. Graphics are used by designers and animators from around the world. The stories are supported by visual material (sketches, animations) made by professionals and can be published (embedded) on other websites (e.g., blogs). Included lessons are done with video tutorials, quizzes, and tips from authors. Allows education and writing texts for novels, songs, poems, screenplays, etc. Registration on the site is required before use. Attending classes and creating stories are free. The courses are divided into learning levels. Downloading or printing a story costs money. The Storybird does not support the enrichment of stories with audio features such as voice or music. Additionally, it does not allow digital history to be stored locally on the user's computer.
- **Book Creator** (https://bookcreator.com/) (accessed on 30 March 2022): Book Creator is designed to teach students by getting them excited about creating their own books

on the topics they are learning about. Students can upload images, choose from emojis, make recordings and videos, and create and then share a finished book they wrote.

- **My Story Maker** (https://www.mystory.co/) (accessed on 30 March 2022): My Storymaker is a free online application that allows users to create their own story by selecting a character and a target. Learners choose characters, sets and objects. The stories are enriched with the interaction of the heroes with each other and with the objects that participate in the story. The story is stored in the form of an interactive book, which can be saved online, read or edited, printed and shared with friends online. The simplest application of storytelling and storytelling for children in the classroom and beyond. It is excellent for children aged 5–12 years. It is possible to separate the story of each child in his own library. Sharing on the Web, Facebook, Twitter, ePub and in movie format. Export and share the story with audio and video on YouTube. Easy and fun user interface. Sharing features are some of the most powerful in the App Store. Teachers and parents alike love this app, creating ePubs, Facebook/Twitter, exporting iBooks and more.

- **My Storybook** (https://www.mystorybook.com) (accessed on 30 March 2022): MyStorybook is a digital online tool that students can use to build and share their own storybooks. The ease of use is a key feature that has made it known to the educational community and shows a preference for it. The user creates the stories from scratch, starting from the cover of the book where he will choose the title, then from the graphics library, he will use characters, images, different backgrounds and designs. The My Storybook tool is an easy-to-use tool for preschoolers aged 4–6 years that guides them step by step in creating their own book. With this tool, children compose a story, and its cover, select images from the graphic library of the tool or from their own archive and are gradually introduced to the concept of illustration and narration. Tool requires registration with an e-mail account and accepts text in Greek. Then, when the story is ready, it is published on the internet, and for a small fee, it can be printed as a regular book.

- **Voki for Foreign Language** (https://l-www.voki.com/) (accessed on 30 March 2022): Voki is an online tool that offers the ability to create digital representations of subjects (avatars) on the Internet with the ability to speak, which can be published on any blog, website or profile and integrated into the activities of various subjects. It is a free online program that belongs to the ten best educational tools of Web 2.0 [25]. The use of Voki in language teaching is considered to be very helpful, as it enables students to practice with different voices (male or female) and with English accents. These familiarization possibilities would not be possible with conventional supervisors and the technological means that a school classroom can have. It is suitable for use by students of all ages. Suitable for creating lesson plans in English.

*2.2. Serious Games—Gamification Tools*

The development of technology could not set aside learning. As a result, teachers and educators are constantly searching for new ways that would trigger the students' interest. Throughout the last decades, the number of digital game-based learning (DGBL) studies has significantly increased, and DGBL seems to leverage traditional instruction and have a significant impact on education [26]. The DGBL effectiveness, nevertheless, varies according to subjects taught or tasks learners are engaged in [27]. Therefore, digital media offer many tools and platforms that can be used for this reason. Serious games and gamification consist of a whole new area in the field of education, coming from the digital media [28].

As the use and availability of games as multimedia student-centered tools in education continue to grow, students' perspectives regarding serious games have become increasingly relevant. Due to the increased popularity of games among people of all ages in their free time, educators have concluded that serious games should also be employed in learning contexts to engage students [29].

Serious games and gamification even though some consider them as being the same, are not. Following, there is an attempt to distinguish them. Serious games and gamification have in common the fact that they both share game elements. However, games incorporate a mixture of game elements, while gamification involves the application of an individual game element. Gamification is a term that originated in the digital media industry. It was firstly used in 2008, nevertheless, its widespread adoption came in the second half of 2010, when several industry players and conferences popularized it [30].

The toolkit of gamified learning is made up of the components of serious games that game designers adapt in order to better learn. When a shared game element taxonomy is used to align them, this link means that existing research on serious games should influence gamification research, and existing research on gamification of learning should inform serious games research [31]. A serious game relies on specific rules and actions that participants must do, the ranking is defined by these rules and actions. Even though both serious games and gamification engage players, the serious game does so through its design, where a total score is dependent on the player's entire behavior or performance in the game.

Gamification, on the other hand, engages users in certain behaviors and does not require this to be implemented within a game. For instance, gamification can occur independently. A game, similarly, usually indicates winners and losers with people vying against one another. Yet, depending on the game features employed, the notion of rivalry may not always have to be present in gamification. Rather, the game components may be used to motivate a user to strive for greater success regardless of the performance of other participants.

Gamification can be used to encourage certain behaviors among individual users, such as submitting material to the platform and therefore to their communities or participating in community events. Examples of these can be found in the preceding sections. Individual behaviors are implicitly rewarded by the platform's recognition of them, for example, using features such as "like" or interest surrounding them. The serious game, on the other hand, is intended to foster collaboration and a sense of community. As a result, there is a synergy between the social game and gamification, where components of the game support the conclusion of the gamification and vice versa, harnessing the capabilities of both serious games and gamification.

A serious game combines a set of rules and actions in a logical manner, with players scoring and competing against one another. The game can be played by solo players or by groups of players, with the latter encouraging teamwork and possibly fostering a sense of community. Gamification, on the other hand, incorporates elements of game design, but is not a game. Gamification can inspire and engage users by allowing them to compete with one another or simply challenge themselves to better their performance [32].

The notion of 'gamification' has appeared as an expression of the pervasiveness of gaming in everyday life. One of the best examples of 'gamification'—or how games are pervading our lives—is the example of serious games, educational gaming as well as games and virtual worlds that are specifically developed for educational purposes reveal the potential of these technologies. A broad definition refers to serious games as computer games that have an educational and learning aspect and are not used just for entertainment purposes. Serious games are currently being used in a range of different contexts [33,34].

When using gamification in the classroom the students become co-designers of the lesson plan, since they can co-decide on the goals and they may attempt an answer numerous times, depending on the rules being set. In the meantime, their progress is visible and the awarding is immediate. Some popular educational serious games and gamification tools:

- **Kahoot** (https://kahoot.it/) (accessed on 30 March 2022): It is a program that allows you to easily construct a question and answer several games. You can design your own quizzes or join one of the many that have already been established and are accessible for a variety of ages and levels. It provides rankings, learning, and entertainment at the same time. It is a free online platform for creating gamification games for students

of all ages and may also be used in special education. It is compatible with all mobile devices. There are two options: The classic mode in which each student uses his own device or the Team mode in which students play in teams using one device. Using the Drops language learning app, users learn a new language.

- **Edmodo** (https://new.edmodo.com/) (accessed on 30 March 2022): Edmodo is the closest thing to a social network for educational purposes, and with the ability to distribute badges to students. Teachers are able to implement the necessary gamification in the classroom. It generates challenges and exercises and awards pins to several pupils, not just the best. It is a free online platform that can be used to create gamification games to encourage students to practice language skills such well as spelling and grammar. This platform may also be used to enhance oral skills of a foreign language which is being taught through conversations. It is a multifunctional tool that can be used by a teacher since it also provides assessment, formative assessment, classroom management, instructional strategies as well as parent communication. It is suitable for students K-12.
- **Baamboozle** (https://www.baamboozle.com/games) (accessed on 30 March 2022): A free website that enables a foreign language teachers to choose between 500,000 available games, covering vocabulary, grammar, tenses, and sentence structure that are made by other teachers. The teachers may also create their own Baamboozle games, adjusting them to their lessons. Baamboozle enables students to compete against each other, as students win points for each correct answer. Its special feature, which students enjoy, is that points can be swapped and stolen from their classmates. Games that are created on this platform can be applied to all ages.
- **Mingoville** (http://www.savivo.com/) (accessed on 30 March 2022): Mingoville is a web-based platform for language learning targeted at primary school learners (ages 9–10). The programme was introduced in Denmark in 2006 as "the world's most comprehensive English language course online for kids of all ages" (Sorensen, 2007). The course has been translated into 31 languages and is sold and marketed worldwide. Mingoville is based on a narrative concept built on the familiar world of the family. The characters of the game are citizens of the simulated world of Mingoville—a city inhabited by flamingos.

### 2.3. Augmented Reality—Virtual Reality Tools

An Augmented Reality (AR) system supplements the real world with virtual (computer-generated) objects that appear to coexist in the same space as the real world [35]. Azuma et al. [36] define an AR system to have the following properties: (1) it combines real and virtual objects in a real environment (2) it runs interactively and in real-time and (3) it registers (aligns) real and virtual objects with each other.

Through AR technology digital content can be seamlessly overlaid and mixed into our perceptions of the real world. Along with the 2D and 3D objects which are expected to be seen, digital assets such as audio and video files, textual information or tactile information may also be incorporated into users' perceptions of the real world. The aforementioned augmentations can serve to aid and enhance individuals' knowledge and understanding of what is going on around them [37].

In the field of education, AR can offer ubiquitous learning. Learners will be able to gain immediate access to a wide range of location-specific information, compiled and provided by a variety of sources. AR learning tools allow students access to capabilities and resources that can dramatically increase the effectiveness of their individual studies. For example, a student who wants to learn a foreign language can use AR simulations to picture where to place the tongue in order to mimic correct pronunciation [36]. Professionals and researchers have worked to incorporate augmented reality into classroom-based learning in areas such as chemistry, mathematics, biology, physics, astronomy, and other K-12 or higher education, as well as augmented books and student manuals [37]. Furthermore, incorporating augmented reality technology into a book can become "magic". Lee [38]

discovered that superimposing 3D rendered models onto books with AR technology allows individuals, particularly young children, to read books in a more engaging and realistic way. This is called "The MagicBook," and it uses both a regular book and a handheld see-through AR device to make people's fantasies of becoming a part of the story a reality.

The main advantages of the use of AR technology in education are that it enhances learning motivation and it increases the student's interest, satisfaction and enjoyment. It also allows learners to learn by doing; it enables the visualization of invisible concepts, events, and abstract concepts. It also allows multi-sensory learning and combines physical and virtual worlds. Furthermore, it facilitates communication between students and lecturers, and it provides collaboration opportunities for students [39,40].

Some AR applications suitable for language educational purposes are the following:

- **AR Flashcards-Animal Alphabet** (https://arflashcards.com/) (accessed on 30 March 2022): AR Flashcards are a way to interact and make Flashcards more entertaining for toddlers and preschoolers. This application is compatible with iOS devices.

- **Mondly AR** (https://www.mondly.com/ar) (accessed on 30 March 2022): Mondly's AR app offers virtual lessons and useful, real-life conversations in 15 languages. A virtual teacher has conversations with the learner, processes the spoken language, and offers instant feedback. You can get instant feedback on your pronunciation and you get the opportunity to practice real conversations—and hear them played back to you in conversation form. With Mondly AR, users get to learn differently. There are life-size animals and other objects that appear in the virtual learning room. This application is suitable for all ages and along with the app a headset, a touchpad and a Gear VR Controller are needed in order to play the game.

- **Catchy Words AR** (https://apps.apple.com/us/app/catchy-words-ar/id1266039244) (accessed on 30 March 2022): Through this application students walk around and catch the letters with the device and solve the word. It is a free application suitable for elementary school students. It is compatible only with iOS 11.0 or later software.

- **Narratorar.com** (https://www.narratorar.com.au/) (accessed on 30 March 2022): Children connect to their written words as the Narrator AR app launches letters off the page in augmented reality (AR). It is an offline, ad-free app suitable for preschoolers and can be downloaded free from AppStore and Google Play.

Virtual Reality takes place within an artificial environment and a participant becomes a part of this artificial world as an immersive or a non-immersive member in contrast to AR. People can interact and manipulate computer-generated objects in a virtual environment with the help of gadgets such as haptic devices. In addition, VR gadgets have influenced VR content and enhanced VR capabilities for better experiences. Smell, wind, sounds, heat, and body movement detection are elements that potentially create VR experiences more realistic and interesting. Virtual Reality (VR) is different from Augmented Reality (AR) in that in VR people are expected to experience a computer-generated virtual environment [41].

A child learns through play from the time he or she is born. The process of learning about the world begins with reaching, touching, gazing, smelling, and tasting whatever the child comes into contact with. The kid begins to identify different properties with different objects through a combination of all senses, and through memorization, is able to build unique categories and concepts from the seemingly disparate and chaotic signals it receives from the world. Perception and activity are essential for learning even in adulthood. VR technology can be used to exploit the link between perception and action for educational reasons. It enables students to experience a variety of scenarios, even ones that are physically difficult to recreate in the classroom. From physics and mathematics to history, archaeology, and cultural heritage, the range of subjects that can be taught is vast. Virtual reality allows users to visualize both the macroscopic and microscopic worlds at a human scale, allowing them to gain an insight that would otherwise be hard to acquire via traditional means [42].

Some indicative applications that use VR technology are the following:

- **House of Languages** (https://www.oculus.com/experiences/gear-vr/1129567930394285/) (accessed on 30 March 2022): Through the House of Languages, someone can learn English, German or Spanish in a fun and creative way. The learner is taught by Mr. Woo, and he can visit the airport, the zoo, the café, and some other places in virtual reality. It is a highly effective way of learning new basic vocabulary. This application is suitable for all ages. Equipment such as a headset and a touchpad are needed to play the game.
- **Immerse Me** (https://immerseme.co) (accessed on 30 March 2022): ImmerseMe is another academic language tool. With this app, you can choose between nine different languages and from over 3000 different scenarios. The languages someone can learn with ImmerseMe are German, Spanish, French, English, Japanese, Chinese, Italian, Greek, and Indonesian. It is classified into three levels: Beginner, Intermediate and Advanced, and is mostly addressed to older students and adults. Equipment needed: Android Cardboard, Vive, Rift or Gear VR.
- **Virtual Speech** (https://virtualspeech.com/courses/english-for-business-vr) (accessed on 30 March 2022): English is the major language for communication in the world of international business. The VirtualSpeech English for Business course focuses on this sector, allowing people from around the world to boost their listening and speaking skills in a business context. VirtualSpeech is focused on improving your communication skills. In other words, their goal is to make you a more confident speaker and a better listener. Its voice analysis technology can analyze your pace of voice and pick up hesitant words, giving you feedback on how to speak more clearly.
- **FluentU** (https://www.fluentu.com/) (accessed on 30 March 2022): FluentU takes authentic videos—like music videos, movie trailers, news and inspiring talks—and turns them into personalized language learning lessons. FluentU's video player supports your students so they can learn on their own. 10 languages and 10,000+ videos that your students will love. Students can access FluentU through the website or iOS/Android mobile apps.

## 3. Educational Tool Comparison—Selection of Language Tools

Therefore, eighteen educational tools were chosen based on the possibility to enable students to practice a variety of language skills independently and also interactively when the keyword "foreign language learning" was entered during our research. Additionally, based on literature suggestions, some educational tools were omitted because they were deemed irrelevant to the research goals, such as those that taught computer programming languages or were exclusively devoted to natural sciences.

Regarding the evaluation criteria selection, numerous frameworks for evaluating educational applications have been developed [43,44]. Among the parameters used to determine efficacy, a few criteria are shared by the majority of frameworks; they include technical characteristics, design, and the app's suitability for its intended use. The factors most commonly highlighted are relevance and authenticity—whether targeted abilities are performed in an authentic format/problem-based learning environment. Additionally, proper navigation, support, accessibility, security, image and sound quality, usability, pricing, feedback, interactivity, content relevance, and instructions are all considered.

In Section 3, we summarize all of the aforementioned educational tools, including comparative tables (Tables 1–3) separately depending on each type of technology (digital storytelling, serious games, AR/VR) and summarize their functionalities. Furthermore, we compare them on the basis of key attributes, as listed below. The fundamental characteristics that enable us to compare educational tools are as follows:

- Relevance and authenticity;
- Proper navigation;
- Accessibility;
- Open-source;
- User input;

- Image and sound quality;
- Platforms;
- Age;
- Multi-language;
- Collaboration;
- Extra features.

**Table 1.** Comparison of storytelling educational tools.

| Educational Toll | Relevance and Authenticity | Proper Navigation | Open Source | User Input | Image and Sound Quality | Platforms | Ages | Multi Language | Collaborative | Extra Features |
|---|---|---|---|---|---|---|---|---|---|---|
| Story Jumper | Medium | ✓ | ✓ | ✓ | Medium | All | 4+ | ✓ | ✓ | Share story |
| Storybird | Medium | ✓ | ✓ | ✓ | High | All | 4+ | | ✓ | Visual material |
| Book Creator | | | 40 books | ✓ | Medium | iOs | 4+ | ✓ | ✓ | Upload media |
| My StoryMaker | Medium | | ✓ | ✓ | Medium | All | 5–12 | | ✓ | Character-based |
| My Storybook | Low | ✓ | ✓ | ✓ | Low | iOs | 4–6 | | ✓ | Print the story |
| Voki | High | ✓ | Only basic | ✓ | Medium | All | 4+ | ✓ | ✓ | 3D animation |
| Kahoot—Drop Language | High | ✓ | ✓ | ✓ | ✓ | All | 4+ | ✓ | ✓ | Drops language tool |
| Edmodo | Medium | ✓ | ✓ | ✓ | Medium | | 4+ | ✓ | ✓ | Social network |
| Mingoville | High | ✓ | Free trial | ✓ | High | All | 4+ | ✓ | ✓ | colourful e-learning environment |
| Mondly kids | High | ✓ | ✓ | ✓ | High | All | 4+ | ✓ | ✓ | Chatbot and speech recognition |

**Table 2.** Comparison of serious games educational tools.

| Educational Toll | Relevance and Authenticity | Proper Navigation | Open Source | User Input | Image and Sound Quality | Platforms | Ages | Multi Language | Collaborative | Extra Features |
|---|---|---|---|---|---|---|---|---|---|---|
| Kahoot—Drop Language | High | ✓ | ✓ | ✓ | ✓ | All | 4+ | ✓ | ✓ | Drops language tool |
| Edmodo | Medium | ✓ | ✓ | ✓ | Medium | | 4+ | ✓ | ✓ | Social network |
| Baamboozle | Medium | | ✓ | ✓ | Medium | | 4+ | ✓ | ✓ | Swap points |
| Mingoville | High | ✓ | Free trial | ✓ | High | All | 4+ | ✓ | ✓ | colourful e-learning environment |
| Mondly kids | High | ✓ | ✓ | ✓ | ✓ | All | 4+ | ✓ | ✓ | Chatbot and speech recognition |

**Table 3.** Comparison of AR/VR educational tools.

| Educational Toll | Relevance and Authenticity | Proper Navigation | Open Source | User Input | Image and Sound Quality | Platforms | Ages | Multi Language | Collaborative | Extra Features |
|---|---|---|---|---|---|---|---|---|---|---|
| AR Flashcards | Medium | ✓ | ✓ | ✓ | Medium | iOs | 12+ | ✓ | | 3D animation |
| Mondly kids | ✓ | ✓ | ✓ | ✓ | ✓ | All | 4+ | ✓ | ✓ | Chatbot and speech recognition |
| Cathy words AR | Medium | ✓ | ✓ | ✓ | Medium | iOs | 6+ | ✓ | | Create your own word list |
| Narrator | Low | ✓ | ✓ | ✓ | High | All | 8+ | | ✓ | Free lesson plans |
| House of LAnguages AR | Medium | | 499€ | ✓ | Low | | 10+ | ✓ | | Headset and touchpad need |
| Immerse Me | Low | | 300 €/year | ✓ | Medium | | 15+ | ✓ | ✓ | Android Cardboard, Vive, Rift or Gear VR needed |
| Virtual Speech | High | ✓ | 45 €/month | ✓ | Medium | | 16+ | ✓ | ✓ | Real-life situations |
| FluentU | High | ✓ | 20 €/month | ✓ | High | All | 15+ | ✓ | ✓ | Audio dialogues are free for offline reading |

As we can see from the tables above, ten out of eighteen of the tools analysed develop digital storytelling content. In education, digital storytelling refers to creating short-term audiovisual work and includes photographs, digital stories, music, recorded narration and video use. Five of the tools out of eighteen include serious games—gamification procedures, and eight out of eighteen support augmented reality or virtual reality modules. Eleven of the eighteen are offered for free in a mobile version, while the rest of them offer a free trial and four are only iOS oriented. Additionally, ten of the eighteen pertain to children under the age of four, four to children between the ages of five and twelve, and four to children aged twelve and beyond. The majority of tools (except four) support multi-language features, and thirteen out of eighteen applications provide users with a wide media collection and collaborative features. Additionally, nine out of eighteen tools are relevant to the foreign language educational procedures and twelve out of eighteen tools have proper navigation. Thus, after comparing the educational tools, we conclude that the most proper educational tools are Mondly Kids and Language Drops-Kahoot, as they meet all of the above criteria.

## 4. Empirical Study at School

The evaluation of an application by its users is critical in its development. The findings drawn regarding the user experience are critical, and a careful interpretation of the results enables teachers to optimize the application's performance. The most generally used evaluation methods include questionnaires, interviews, and user observations [28,45,46].

In the course of the research process and in order to collect the necessary empirical material, we chose the techniques that we considered to best meet the requirements and the nature of empirical research. Observation as an evaluation methodology was used as the first method of collecting our data. The use of direct observation as a method of data collection by teachers has particular advantages, especially in relation to the evaluation of complex teaching and learning processes. Techniques such as notes, observation lists, photography and video recording were used to record the findings during the observation. In combination with other data mining techniques, such as the questionnaire, it can be a useful and powerful tool for gathering information in English classroom teaching.

As a second tool for evaluating our tools, therefore, an online questionnaire (Appendix A) was used through Google Forms, according to which students had the freedom to answer as completely and spontaneously as they could. For the younger students in the Kindergarten who have not yet acquired the skills of reading and writing, the answers were recorded by the Kindergarten teacher-researcher. The questionnaire is a data collection tool, in which the research subject is asked to answer in writing a series of pre-designed questions on a topic. The usefulness of the questionnaire depends mainly on the quality of the questions, as it is very difficult or even impossible for the researcher to ask clarifying questions afterwards. Its most important advantage is that it can be answered anonymously, which makes it easier for respondents to be completely honest. For the evaluation, end-users (students) were asked to test the teaching scenarios through the application; teachers observed their reactions during the process and the interaction with the application. Both quantitative and qualitative methods were used to evaluate the applications. Observation of the students' reactions (mainly the younger age groups) by the teacher as well as the completion of questionnaires by the students were both combined in order to draw conclusions about the effectiveness of the applications. Then, the students answered the questionnaire. Students either participated in groups (school class) or individually. The sample consists of 60 students. Specifically, 20 students aged 4–6 years, 26 students aged 7–12 years, 6 students aged 13–15 years and 8 students aged 18 years and over. The teachers who participated were three. One Kindergarten Teacher, one English teacher and one Spanish teacher. The implementation of the educational scenario lasted 5 days for each age level. The activities that took place were about nutrition and animals and were carried out with the digital tools Mondly Kids and Language Drops-Kahoot. The selection of the digital tools was made based on the curriculum of each level, the age and the abilities of the students. The methodological approaches used were the interdisciplinary and experiential approaches to knowledge, as well as the ability for collaboration and self-action. Therefore, for evaluating the two digital tools Mondly Kids and Language Drops-Kahoot, we created three different groups of students. The students were separated in three different age groups in order to study the effects of the applications on each group separately, since the applications were of different levels according to their age:

- An age group of 4–6 years was done in a real classroom in a section of Public Kindergarten Kefalonia and the subjects studied were a total of 19. The children were already familiar with the use of the computer, with similar software and digital tools of preschool age.
- An age group of primary school students to supplement the vocabulary of the "animals" unit, which was completed within one week. We divided students into groups of 2–3, and students played with the apps in turns.
- There is an age group of young learners or beginner learners (10+ age) of Spanish as a foreign language.

For the first group, the use and development of digital tools to enhance the teaching of English to students aged 4–6 are more than obvious in the educational process. It makes learning more accessible and attractive while at the same time promoting collaboration and the interdisciplinary approach to knowledge.

Their application showed that children aged 4–6 years can interact with new technologies with the teacher's support (in downloading games on tablets, in the steps they had to follow). The children acquired a positive attitude towards digital media, developed communication skills and at the same time increased their digital knowledge and skills. Both Mondly Kids and Language Drops proved through the students' answers to the questionnaire that they are age-appropriate applications for children.

The dynamics of images, colors in both applications, and music and movement made learning more accessible and attractive at these young ages. The pictures and graphics of the games helped the students significantly in enriching their English vocabulary and strengthened their imagination and creativity. The children were thrilled with the possibilities of the games, cooperated, interacted and got acquainted very quickly with the choices

of the toys. They did not face any particular difficulties in using them. The young learners (4–6-year-old students) did not seem to face particular difficulties in using the applications since after they were shown by the teacher the way they were played, they immediately started playing the games without the need for further explanations. In terms of digital skills, some students were more familiar; others found it a little difficult to get acquainted, while some tried to solve possible problems independently. When they failed, they asked for the help of their kindergarten teacher or classmates. Regarding cooperation, it was observed that the students managed to cooperate in their groups to the greatest extent.

By combining entertainment with learning, students acquired new concepts related to animals, and food, and developed attitudes and skills through the digital stimuli they gave each time. In addition to the students' cognitive skills, their socio-cognitive skills were also enhanced. We believe that the way the action developed evolved into an interesting, positive and useful experience through which children learned, created, had fun and gained new knowledge (Figures 1 and 2).

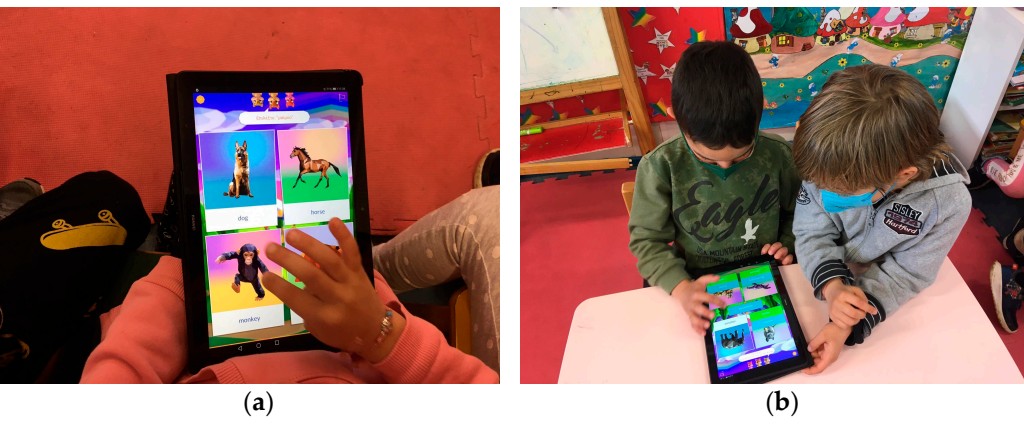

(**a**)　　　　　　　　　　　(**b**)

**Figure 1.** (**a**,**b**). Mondly Kids school evaluation—first group user observation.

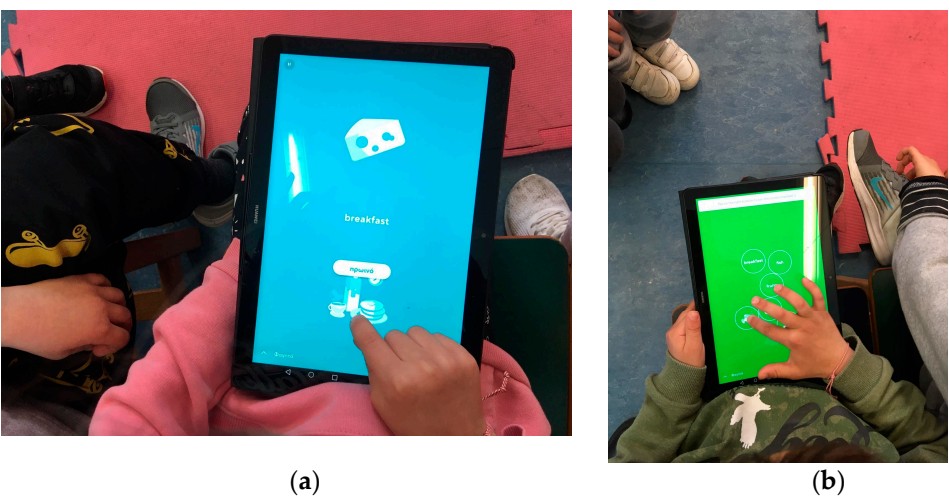

(**a**)　　　　　　　　　　　(**b**)

**Figure 2.** (**a**,**b**). Language Drop-Kahoot school evaluation—first group user observation.

For the second group, we noticed that the students had a positive reaction towards these apps. At the same time, they consolidated the vocabulary to be much more accessible than with traditional methods.

Throughout the game, the children showed a tremendous deal of enthusiasm, delight, and enjoyment, which they conveyed in various ways. They often high-fived and did the victory sign. At the same time, they shouted "yes" while clenching their fist when giving the correct answer. Additionally, children of the same group cooperated and helped each other.

Another useful finding is that students who were weak or others with learning disabilities were welcomed into the teams with ease and were assisted during the games. The young learners enjoyed the colors, graphics and sound effects. They looked forward to the next lesson to use the applications again and review the vocabulary they had already learned.

This type of application will undoubtedly be included in the weekly program since the interactive activities, and word games assist young children in learning new words. They related words to illustrations; they used drag-and-drop functionality and multiple-choice activities, and all the above-assisted learners to remember the words faster. Additionally, high-quality audio helped learners to adopt a better pronunciation. All in all, gamified lessons can be used to keep students interested in the learning process. The applications of this kind should be included in the weekly schedule of the subject English language (Figures 3 and 4).

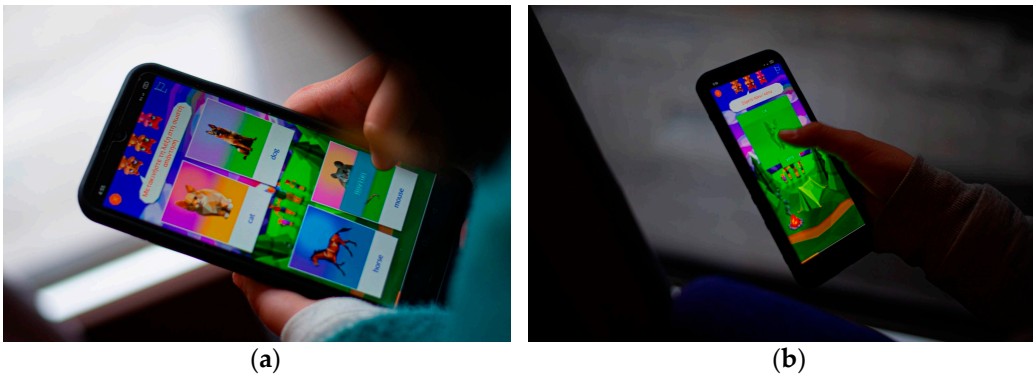

(**a**)                                                    (**b**)

**Figure 3.** (**a**,**b**). Mondly Kids school evaluation—second group user observation.

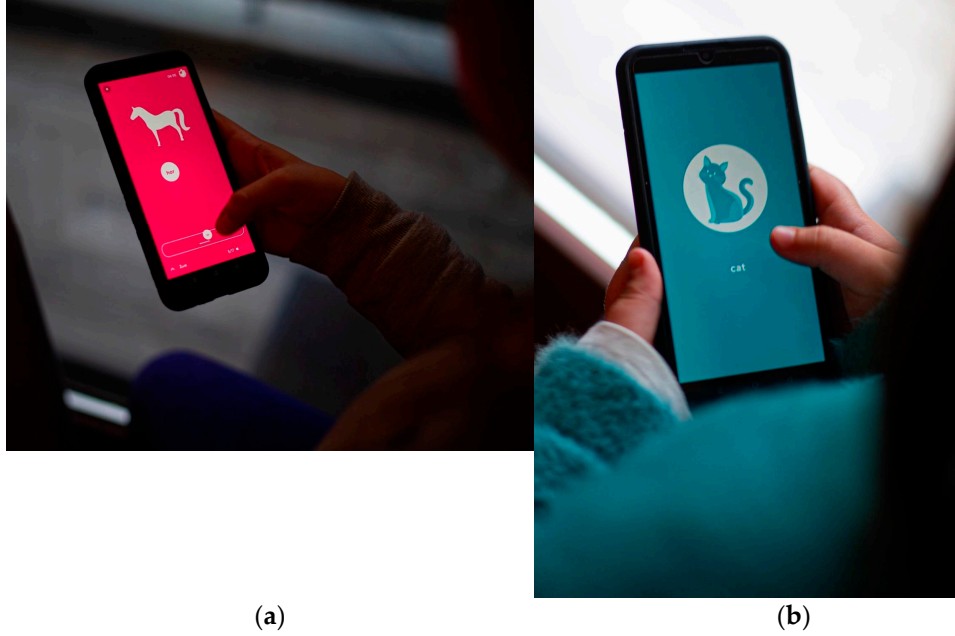

(**a**)                                                    (**b**)

**Figure 4.** (**a**,**b**). Language Drop-Kahoot school evaluation—second group user observation.

For the third group, the Mondly Kids app contains a number of vocabulary games addressed to young learners or beginner learners of Spanish as a foreign language. It offers a wide variety of categories of games to choose from. The Mondly Kids app was used to reinforce the learning of the new vocabulary that was presented at the beginning of the lesson. The unit studied this week was about animals, so the app selected the relevant

topic. The students installed the application on their phones, selecting their mother tongue and the language they were learning (Spanish). Then, they set the category "animals" and started playing various games such as drag and drop or matching games.

The learners were excited by the graphics and the interface of the games. While they were playing, we observed their satisfaction every time they achieved a correct answer or completed a task. Additionally, while they were playing, they even glimpsed at their classmates' smartphones to check whether the others were at the same level as them or if they were left behind. They tried to give the correct answers as fast as possible to finish first. When the game ended, the students asked if they could play another round, and they were looking forward to next week's lesson, which would be devoted to another topic and, therefore, another game.

We used the Language Drops app to supplement the learning of the vocabulary encountered in the "food" unit of the students' coursebook. The students downloaded the app and played games on the relevant topic. They had to match the written word with the appropriate picture. They also heard the word which was displayed, and they repeated it. The students enjoyed playing the game and smiled every time they completed a task.

Adults are also learning Spanish as a foreign language. They downloaded the app and played the games containing the vocabulary about the working environment previously presented during the lesson.

The learners found the interface user-friendly and commented that this interactive app helped them memorize the new vocabulary quickly and was fun. They also found the app handy since they would soon revise the language taught anytime or anywhere, they wished to (Figures 5 and 6).

After using the apps, all students were asked to fill out the questionnaire. In the case of young ones, they were helped by the teacher who read the questions for them (see responses in Figures 7–10).

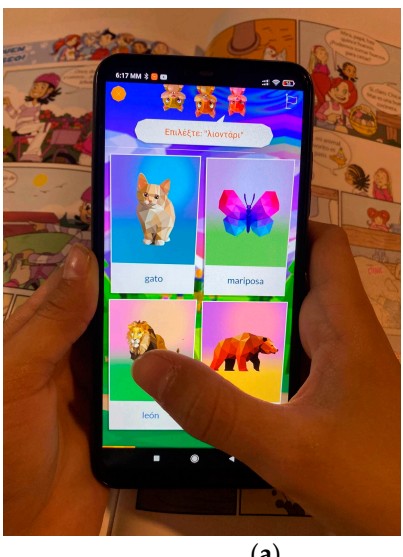 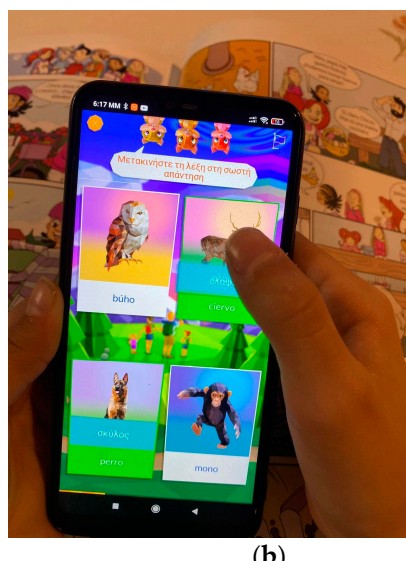

(**a**)  (**b**)

**Figure 5.** (**a**,**b**). Mondly Kids school evaluation—third group user observation.

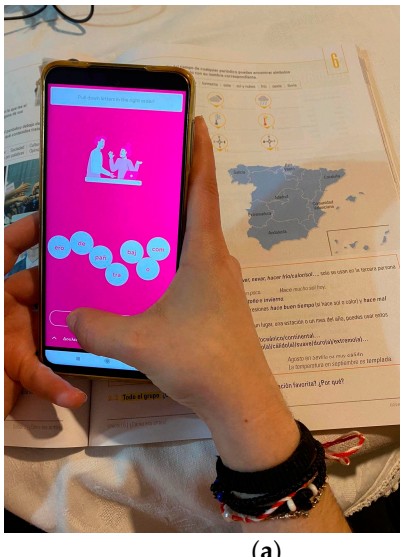

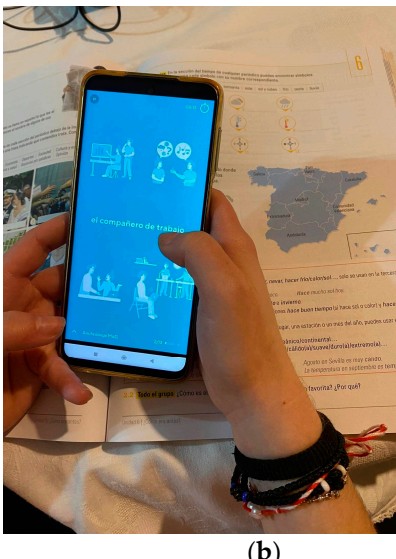

| (**a**) | (**b**) |

**Figure 6.** (**a**,**b**). Language Drop-Kahoot school evaluation—third group user observation.

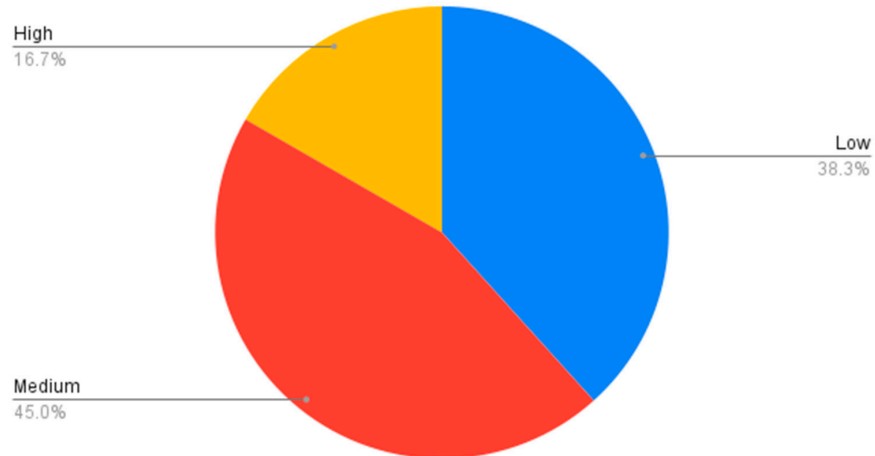

**Figure 7.** Familiarity with the use of new technologies.

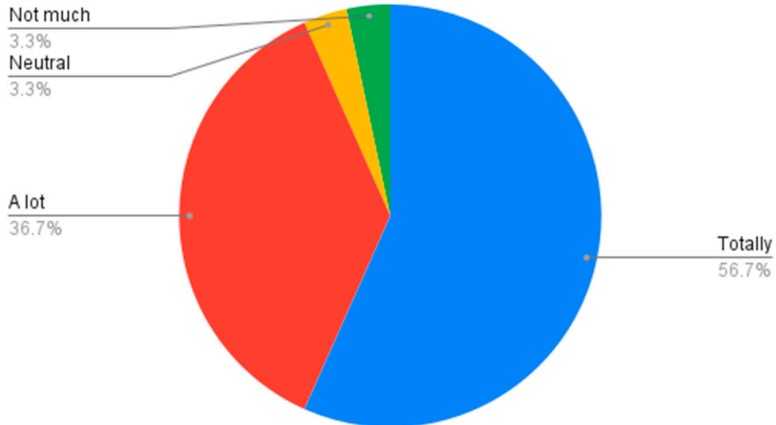

**Figure 8.** Willingness to learn.

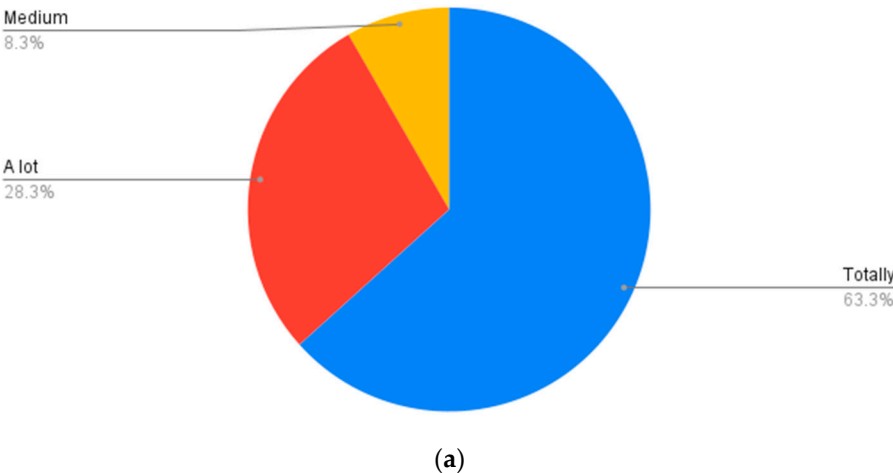

(**a**)

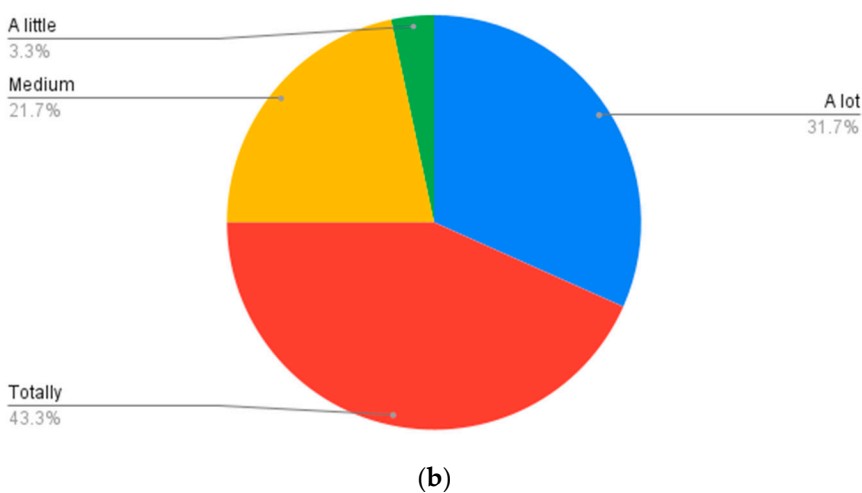

(**b**)

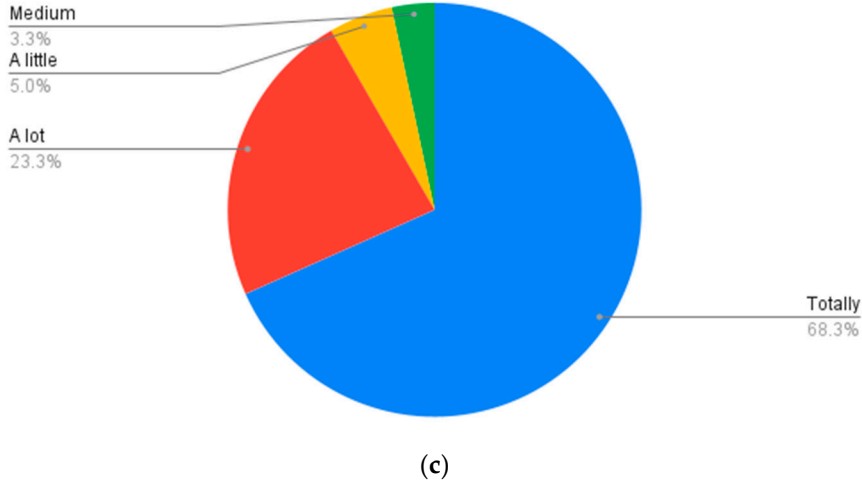

(**c**)

**Figure 9.** (**a–c**). Evaluation of the audiovisual material of the application (images, videos, sound effects).

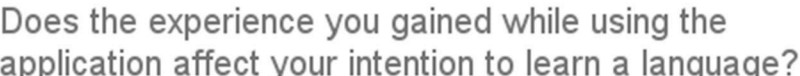

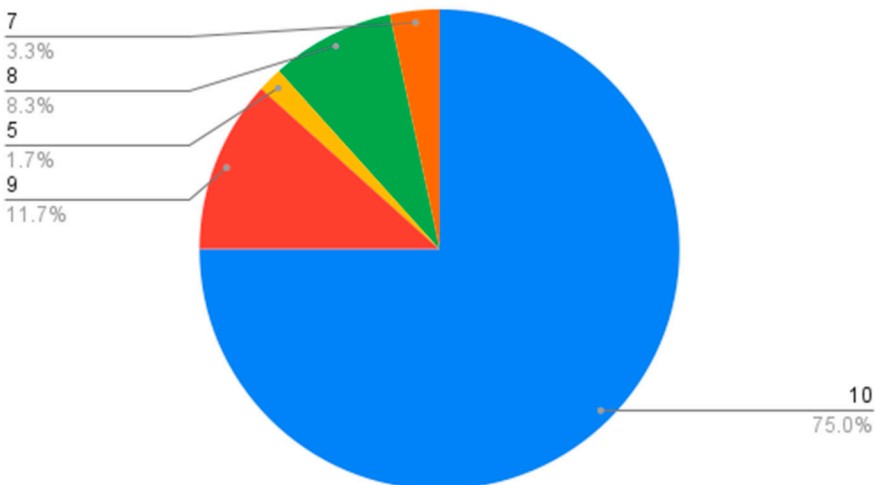

**Figure 10.** Intention to learn a language after using both of the applications (on a scale of 1–10 where 1 = Definitely no and 10 = Definitely yes).

## 5. Data Analysis and Results

This part analyses the assessment process's data, presenting the most pertinent information for each group. Figures 7–10 illustrate the graph analysis, while Table 4 illustrates the mean and standard deviation data analysis.

As a result of collecting and analyzing the responses and comments, we arrived at the following:

- The majority of students perceived the experience as a positive educational opportunity to gain new knowledge about language learning.
- Students gave the Language apps an eight on a scale of 1 to 10 for their ability to entice them to continue using them after two minutes.
- Most participants expressed a desire for social media interaction while stating that the interactive content creation process aided in language learning promotion.
- On a scale of 1 to 10, the students gave the applications an 8 for design.
- Regarding the audiovisual material (images, videos, and sound effects), it was delightful, enjoyable, unique and collaborative.
- The results indicate that the applications provide significantly more features than other educational applications, significantly increase learners' willingness to learn, and provide a fully authentic experience.
- Additionally, students agree that the applications do not take up much space in the device's storage memory, do not significantly reduce the device's battery life, and the mobile device's screen size is adequate.
- Installing the application was more straightforward than using it.
- On a scale of 1 to 10, students scored nearly 10 for their ability to comprehend the language procedure and its effect on their desire to learn a new language.
- Some additional features were identified, such as the need for other donation games, additional activities to complete, a more interactive experience, and different topics.

**Table 4.** Mean and standard deviation analysis.

| Question | Scale | Mean | Standard Deviation (SD) | Variance |
|---|---|---|---|---|
| Do you like experimenting with new applications? | 1–10 | 8.75 | 1.41 | 1.98 |
| How appealing do you find the design of the application? | 1–10 | 9.41 | 0.85 | 0.72 |
| How would you rate each of the following services of the application? [Graphics] | 1–5 | 4.53 | 0.62 | 0.38 |
| How would you rate each of the following services of the application? [Interaction] | 1–5 | 4.33 | 0.68 | 0.47 |
| How would you rate each of the following services of the application? [Collaboration] | 1–5 | 3.90 | 0.82 | 0.67 |
| How would you rate each of the following services of the application? [Video] | 1–5 | 4.10 | 0.84 | 0.70 |
| Did the app help you understand the language as described during the trial? | 1–10 | 9.48 | 0.96 | 0.93 |
| Would you recommend to a friend to buy the educational app? | 1–10 | 9.13 | 1.27 | 1.61 |
| Does the experience you gained while using the application affect your intention to learn a language? | 1–10 | 9.53 | 0.98 | 0.97 |

## 6. Conclusions

This study aims to provide learning scenarios based on two mobile learning apps that enhance language acquisition through engaging, interactive settings that are better tailored to children's ages. After analyzing educational tools using a variety of factors, we find that Mondly Kids and Language Drops-Kahoot are the most appropriate instructional resources. We constructed three distinct groups of students based on this assumption: a group of kids aged 4–6 years in a genuine English learning classroom, a group of primary school students learning the English language, and a group of young learners or starting learners (10+ age) of Spanish as a foreign language. The assessment procedure's findings indicate that mobile language learning augments children's experiences and boosts their desire to learn a new language. Additionally, mobile applications can help students enhance their speaking ability and critical thinking skills throughout a language learning session.

The contribution of the present study to the educational process but also to science in general lies in the following points. First, the use of digital tools in English language teaching is a very important tool at all levels of education, as it has been shown to increase oral and written speaking skills, while enhancing the critical thinking, analysis and information skills of all of the students. Secondly, in this context of the creative use of new technological tools, digital games are increasingly strengthening their place in the educational process and are a new means of teaching and cultivating the English language to children of all ages and more of preschool age who have not yet mastered the skills of reading and writing. Digital games help to acquire more ICT skills, actively involve students, encourage interpersonal communication and collaboration among students, create learning motivation, can be combined with the involvement of various cognitive objects and in the development of all the skills mentioned in the various learning theories, such as collaborative learning, creativity and innovation. They can be used either at the beginning of the lesson to mobilize and arouse the interest of those involved or as a bridge between pre-existing knowledge and new material. The dynamics of images and photographs in combination with music and movement make learning even more engaging. Thirdly and more important is the fact that no similar study has been conducted in Greece to study the use of these specific digital tools in such a wide range of age groups. Closing and taking into account all the new trends in the educational process, we could say that the role of new digital tools in

English language teaching is crucial, bringing many innovations and upgrading the quality of educational work.

Future studies might examine how self-access learning can be incorporated into mobile apps. Students determine what they will learn, how they will learn it, and how they will judge their own progress. Students may choose when and where to learn with self-access learning, which uses internet resources. Another possible recommendation is to focus on the teacher's worry about the usage of mobile gadgets during class. Teachers must overcome obstacles associated with adopting technology for successful language instruction.

**Author Contributions:** Conceptualization, M.K. and G.H.; methodology, M.K.; software, E.L., A.L. and G.T.; validation, M.K.; formal analysis, E.L., A.L. and G.T.; investigation, M.K.; resources, E.L., A.L. and G.T.; data curation, M.K.; writing—original draft preparation, E.L., A.L. and G.T.; writing—review and editing, M.K.; visualization, M.K. and G.H.; supervision, M.K. and G.H.; project administration, M.K. and G.H.; funding acquisition, M.K and G.H. All authors have read and agreed to the published version of the manuscript.

**Funding:** This research was funded by Ionian University, Department of Digital Media and Communications, research funds.

**Institutional Review Board Statement:** Not applicable.

**Informed Consent Statement:** Not applicable.

**Data Availability Statement:** Not applicable.

**Conflicts of Interest:** The authors declare no conflict of interest.

## Appendix A. Questionnaire [1.2]

The following anonymous questionnaire has been used in this research. All participants gave their consent to processing their responses for research-related tasks, in accordance with national and European data privacy regulations.

*Appendix A.1. Socio-Demographic Characteristics*

**Gender**
(i) Male (ii) Female
**Age**
(i) 4–6 (ii) 7–12 (iii) 13–15 (iv) 16–18 (v) 18+
**Residence**
… … … … …
**Familiarity with the use of new technologies**
(i) Low (ii) Medium (iii) High (iv) Other
**What kind of mobile device do you use?**
(i) Smartphone Android (ii) Tablet Android (iii) iPhone (iv) iPad (v) Other
**Do you have at least one social media account?**
(i) Yes (ii) No (iii) Other

*Appendix A.2. User Experience*

**Have you used any digital educational applications on your mobile phone?**
(i) Yes (ii) No (iii) Other
**Do you frequently use educational tools—apps?**
(i) Not at all (ii) Rarely (iii) Often (iv) Very often (v) Other
**Do you like experimenting with new applications?**
(i) 1 = Not at all (ii) 10 = Very much (1–10 linear scale)
**If you haven't used any educational tools, what is the reason?**
(i) They are not easy to use (ii) It didn't occur (iii) There is not enough material relevant to my interests (iv) Other
**How appealing do you find the design of the application?**

(i) 1 = Not at all (ii) 10 = Very much (1–10 linear scale)

**Please evaluate the AUDIOVISUAL material of the application (images, videos, sound effects, virtual tour, etc.):**

(i) Is it fun? (ii) Is it unique? (iii) Is it pleasant? (iv) Is it collaborative? (multiple choice grid)

**Please evaluate the application based on the following:**

(i) It offers more features than other educational applications that I have tried or heard (ii) It increases the willingness to learn (iii) It offers an authentic experience (multiple choice grid)

**Please tell us how much you agree with the following:**

(i) The personal data collected through the application is safe (ii) The application does not consume much space in the storage memory of the device (iii) The application does not significantly reduce the battery life of the device (iv) The screen size of the mobile device was appropriate/sufficient (multiple choice grid)

**How would you rate each of the following services of the application?**

(i) Graphics (ii) Interaction (iii) Collaboration (iv) Video (multiple choice grid)

**Please let us know how easy it was:**

(i) To install the application on your device (ii) To use the application (multiple choice grid)

**Did the app help you understand the language as described during the trial?**

(i) 1 = Not at all (ii) 10 = Very much (1–10 linear scale)

**Would you recommend to a friend to buy the educational app?**

(i) 1 = Definitely no (ii) 10 = Definitely yes (1–10 linear scale)

**Does the experience you gained while using the application affect your intention to learn a language?**

(i) 1 = Definitely no (ii) 10 = Definitely yes (1–10 linear scale)

**What extra features would you like in an educational app?**

… … … ..

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
