# Peer review of "An Exploratory Study of Mobile-Based Scenarios for Foreign Language Teaching in Early Childhood"

_education, doi:10.3390/educsci12050306_

Round 1
Reviewer 1 Report
Dear author,
Thank you very much for uploading your article for consideration to Education Sciences. This article is very interesting and it can be published after the revision. It is necessary to provide modifications according to the following comments:
- Section No. 2 – systematic review: please provide an extended point of view on this issue and in the field of e-learning and LMS education platform for elementary skills testing, e.g. Bradley, N., Jadeski, L., Newton, G., Ritchie, K., Merrett, S., & Bettger, W. (2013). The Use of a Learning Management System (LMS) to Serve as the Virtual Common Space of a Network for the Scholarship of Teaching and Learning (SoTL) in an Academic Department. Education Sciences, 3(2), 136-146.; Tunmibi, S., Aregbesola, A., Adejobi, P., & Ibrahim, O. (2015). Impact of E-Learning and Digitalization in Primary and Secondary Schools. Journal of education and practice, 6(17), 53-58.; Husár, J., & Dupláková, D. (2016). Evaluation of foreign languages teaching in LMS conditions by facility and discrimination index. TEM Journal, 5(1), 44.; Mitaľ, D., Dupláková, D., Duplák, J., Mitaľová, Z., & Radchenko, S. (2021). Implementation of Industry 4.0 Using E-learning and M-learning Approaches in Technically-Oriented Education. TEM Journal, 10, 368-375.; Sáiz-Manzanares, M. C., Marticorena-Sánchez, R., Muñoz-Rujas, N., Rodríguez-Arribas, S., Escolar-Llamazares, M. C., Alonso-Santander, N., ... & Mercado-Val, E. I. (2021). Teaching and learning styles on moodle: An analysis of the effectiveness of using stem and non-stem qualifications from a gender perspective. Sustainability, 13(3), 1166.; Töröková, M., Dupláková, D., Török, J., & Duplák, J. (2021). Application Use of Augmented Reality in the Educational Process.; Ezaldeen, H., Misra, R., Bisoy, S. K., Alatrash, R., & Priyadarshini, R. (2022). A hybrid E-learning recommendation integrating adaptive profiling and sentiment analysis. Journal of Web Semantics, 72, 100700, etc.
- Section No. 3 - provide information on how you selected the educational tools (methodology, keys, suggestions in literature; and provide information on how you selected the evaluation criteria of education tools selection and criteria from table 1 selection (storytelling, online etc.)
- Section No. 4 – the questions of the questionnaire are missing; the graphical interpretation of the result is very bad; figure 14 and figure 15 – description of the axis is missing; the graphical interpretation of questionnaire results is necessary to rework according to the scientific methodology for statistical data interpretation
- Conclusion – please provide information about the contribution of your research to science and practice
- Please provide your opinion on education depending on building a higher dependency on early childhood habits to use digital technologies
Author Response
Cover Letter
Response to Reviewers’ and Editor’s Comments
Dear Editor,
Thank you for the opportunity to revise our manuscript “An exploratory study of mobile-based scenarios for foreign language teaching in early childhood”. We appreciate the careful review and constructive suggestions. It is our belief that the manuscript is substantially improved after making the suggested edits, highlighted within the document by using coloured text.
Reviewer #1
R1#1: Section No. 2 – systematic review: please provide an extended point of view on this issue and in the field of e-learning and LMS education platform for elementary skills testing
The aforementioned changes have been addressed in the revised manuscript. More specifically, we update the systematic review part with additional citations (5,6,7,8,9,10, 11).
R1#2: Section No. 3 - provide information on how you selected the educational tools (methodology, keys, suggestions in literature; and provide information on how you selected the evaluation criteria of education tools selection and criteria from table 1 selection (storytelling, online etc.)
The aforementioned changes have been addressed in the revised manuscript. More specifically, we update Section 2 with additional citations (43, 44, 45).
R1#3: Section No. 4 – the questions of the questionnaire are missing; the graphical interpretation of the result is very bad; figure 14 and figure 15 – description of the axis is missing; the graphical interpretation of questionnaire results is necessary to rework according to the scientific methodology for statistical data interpretation
The aforementioned changes have been addressed in the revised manuscript. More specifically, Figures 14 and 15 were modified, and the questionnaire was included in the Appendix.
R1#4: Conclusion – please provide information about the contribution of your research to science and practice
The aforementioned changes have been addressed in the revised manuscript.
R1#4: Please provide your opinion on education depending on building a higher dependency on early childhood habits to use digital technologies.
The aforementioned changes have been addressed in the revised manuscript.
We thank R1 for his response.

Reviewer 2 Report
Although the idea underlying this piece of research is interesting, I believe it needs a major revision and restructuring. The major problem observed is that it seems to be two papers in one. The first part (‘systematic’ review) is based on an account of different types of tools related with different technologies, with scarce reference to previous scientific works and results, whereas the second part (‘Empirical study’) needs to be restructured in a more academic form (method, objectives, data results and analysis, etc.)
ABSTRACT. The abstract is not informative enough. It should include the most relevant aspects of the research paper (context, procedure, conclusions, etc). Is this a systematic review of several tools or an empirical study about two tools?
- INTRO and REVIEW section (Literature review). Some statements are not proven, for example
- ‘According to recent research, mobile learning can be both intrinsically and extrinsically motivating and promote cooperation and collaborative work’ Recent research? Which one?
There is a rich body of literature today about the use of some of the tools mentioned by the authors (but not used in their research), for example:
- Statements: ‘They develop literacy skills through designing, writing, and presenting their digital stories. Their critical thinking and skills such as collaboration, communication, creativity and innovation are cultivated.’ (p.2); and ‘Lack of technological equipment, lack of teacher training, technical difficulties are other obstacles that prevent teachers and students from getting involved in creating multimodal stories and making use of technology’ (p.3)
Where are the references or previous work? Check for example:
Nair, V., & Yunus, M. M. (2021). A systematic review of digital storytelling in improving speaking skills. Sustainability, 13(17), 9829.
Belda-Medina, J. (2022). Promoting inclusiveness, creativity and critical thinking through digital storytelling among EFL teacher candidates. International Journal of Inclusive Education, 26(2), 109-123.
- About Storyjumper, Storybird, Book Creator and all the other tools. Check also previous works such as:
Ezeh, C. (2021). A Comparison of Storyjumper with Book Creator, and Storybird for Multimodal Storytelling. Ezeh, C.(2020). A comparison of Storyjumper with Book Creator and Storybird for multimodal storytelling. TESL-EJ, 24(1), 1-9.
Kazazoglu, S., & Bilir, S. (2021). Digital Storytelling in L2 Writing: The Effectiveness of" Storybird Web 2.0 Tool". Turkish Online Journal of Educational Technology-TOJET, 20(2), 44-50.
The authors include all the tools in Table 1 but they mix different types of technologies which have been designed with very different purposes such as AR development kits (immersive technologies), DST tools (storytelling), games, clickers (just-in-time), etc. I suggest to restructure the first part, include a comparative table separately depending on each type of tool or SDK (digital storytelling, games, AR, clickers, etc) and summarize their functionalities. Most importantly, include some previous works about the benefits and limitations of using such tools in the classroom with children. In other words, I believe the first part of the paper lacks academic entity and could be considered a mere account of different tools related with different types of technologies. The promotional information provided in the first part can be easily found on the Internet, different tech experts usually give account of the pros and cons of each tool but this is definitely a more commercial approach. For a research paper, a more academic approach might be needed such as the one proposed above, including references to previous works. As an expert who has used and published on most of these tools, I am not sure if a general table comparing very different tools with different purposes could be useful.
- REVIEW SECTION. Please, explain what is this classification based on?:
‘The fundamental characteristics that enable us to compare educational tools are as follows:’ (p. 9)
As previously explained, I believe table 1 is too large and general as it is comparing different types of technologies. This table could be broken down into several tables related with each type of technology.
4. REVIEW SECTION. The following statement can be misleading: ‘As we can see from the table above, all of the tools analysed develop digital storytelling content’ What do the authors understand by ‘Digital Storytelling’?
5. EMPIRICAL STUDY. This second part of the paper should be restructured to include all the different sections (context, method, data results, etc). There are several statements in this section which require further clarification. For example:
‘For the evaluation, end-users (students) were asked to test the teaching scenarios through the application; teachers observed their reactions during the process and the interaction with the application.’ (p. 10)
Please, explain the research context before (students, teachers, setting, level, etc) How many students? How many teachers?
‘Then, the students answered the questionnaire’ (p. 10)
Which one? Please, provide more details about the instrument.
How did the teachers ‘observe’ the student interaction with technology? What tool or method did they use in their observation?
The authors should explain why they created three groups as they seem to be very different? ‘An age group of 4-6 years’ , ‘An age group of primary school students’, and ‘an age group of young learners or beginner learners (10+ age)’. What is the purpose and rationale behind?
6. EMPIRICAL STUDY. Some statements are difficult to believe, for example:
‘They did not face any particular difficulties in using them’ (p. 11) referring to the first group (4-6 y.ol). How do the authors know that the small children did not face any difficulty?
7. EMPIRICAL STUDY. The authors seem to adopt sometimes an overenthusiastic view of their results, which could be based on a biased approach. For example ‘For the second group, we were delighted to find out that the students were enthusiastic about these apps.’ (p. 12) And also in the statement ‘Another essential and extremely gratifying finding is that students who were weak’ (p. 12)
8. METHOD. I believe the authors should definitely include one specific section about context and research participants, method, instruments, etc before the results. There is a scarcity of details which are included under the section ‘empirical study’. But this section apparently combines method and context with research analysis, so the reader may be confused about some results as the data is not sequentially explained.
The research procedure is not clearly explained:
‘In the end, to complete the evaluation process, the students filled out a questionnaire’ (p. 14)
All participants completed the questionnaire? Small children (4-6 y.o.) too? How did they complete the questionnaire? Which one (use an Appendix)?
9. IMAGES. I believe some images may not be needed as they do not convey any relevant data about the research.
10. DATA ANALYSIS. There is no specific section about this, it is included in the empirical study but the data should not be only based on graphs (Google forms). They should also contain the reliability of the instrument (questionnaire) for each group, the SD, Means, etc.
And better explain some of these graphs such as why did the authors use a 10-point scale and what ‘type’ of technology does each graph refer to (Mondly, Drops), for example Fig. 15?
11. CONCLUSIONS. The authors should separate the data results from the conclusions. The result analysis should be related to previous works as there is no reference to previous literature after the introductory section. The conclusions are very general and not specifically related to the digital tools the authors allegedly used in their research.
12. The authors should state some research limitations.
Some minor English problems observed:
‘Students point to the letters thanks’ (p. 7) ‘thanks’
‘It’s a mandatory investment’ (p. 8) no contraction
‘The educational tools described above, enable students’ (p. 8) no comma
Author Response
Cover Letter
Response to Reviewers’ and Editor’s Comments
Dear Editor,
Thank you for the opportunity to revise our manuscript “An exploratory study of mobile-based scenarios for foreign language teaching in early childhood”. We appreciate the careful review and constructive suggestions. It is our belief that the manuscript is substantially improved after making the suggested edits, highlighted within the document by using coloured text.
Reviewer #2
R2#1: The abstract is not informative enough. It should include the most relevant aspects of the research paper (context, procedure, conclusions, etc). Is this a systematic review of several tools or an empirical study about two tools?
The aforementioned changes have been addressed in the revised manuscript. More specifically, we update the systematic review part with additional text at the last paragraph.
R2#2: Some statements are not proven, for example, ‘According to recent research, mobile learning can be both intrinsically and extrinsically motivating and promote cooperation and collaborative work’ Recent research? Which one? There is a rich body of literature today about the use of some of the tools mentioned by the authors (but not used in their research). Where are the references or previous work?
The aforementioned changes have been addressed in the revised manuscript. More specifically, we update the systematic review part with additional citations (18-19).
R2#3: The authors include all the tools in Table 1 but they mix different types of technologies that have been designed with very different purposes such as AR development kits (immersive technologies), DST tools (storytelling), games, clickers (just-in-time), etc. I suggest restructuring the first part, including a comparative table separately depending on each type of tool or SDK (digital storytelling, games, AR, clickers, etc) and summarising their functionalities. Most importantly, include some previous works about the benefits and limitations of using such tools in the classroom with children. In other words, I believe the first part of the paper lacks an academic entity and could be considered a mere account of different tools related to different types of technologies. The promotional information provided in the first part can be easily found on the Internet, different tech experts usually give an account of the pros and cons of each tool but this is definitely a more commercial approach. For a research paper, a more academic approach might be needed such as the one proposed above, including references to previous works. As an expert who has used and published on most of these tools, I am not sure if a general table comparing very different tools with different purposes could be useful.
The aforementioned changes have been addressed in the revised manuscript. More specifically, Table 1 is broken down into several tables (3 tables regarding storytelling, serious games and AR/VR technology). Also, previous works added regarding the benefits and limitations of using such tools in the classroom with children (5,6,7,8,9,10, 11).
R2#4: Please, explain what this classification is based on?: ‘The fundamental characteristics that enable us to compare educational tools are as follows:’ (p. 9)
The aforementioned changes have been addressed in the revised manuscript. More specifically, we update Section 2 with additional citations (43, 44, 45).
R2#5: As previously explained, I believe table 1 is too large and general as it is comparing different types of technologies. This table could be broken down into several tables related to each type of technology.
The aforementioned changes have been addressed in the revised manuscript. More specifically, Table 1 is broken down into several tables (3 tables regarding storytelling, serious games and AR/VR technology). Also, previous works added regarding the benefits and limitations of using such tools in the classroom with children (5,6,7,8,9,10, 11).
R2#6: REVIEW SECTION. The following statement can be misleading: ‘As we can see from the table above, all of the tools analysed develop digital storytelling content’ What do the authors understand by ‘Digital Storytelling’?
A small paragraph regarding the aforementioned issue has been addressed in the revised manuscript .
R2#7: This second part of the paper should be restructured to include all the different sections (context, method, data results, etc). There are several statements in this section which require further clarification. Please, explain the research context before (students, teachers, setting, level, etc) How many students? How many teachers? How did the teachers ‘observe’ the student interaction with technology? What tool or method did they use in their observation?
A small paragraph regarding the aforementioned issue has been addressed in the revised manuscript at the empirical study section.
R2#8: The authors should explain why they created three groups as they seem to be very different? ‘An age group of 4-6 years’ , ‘An age group of primary school students’, and ‘an age group of young learners or beginner learners (10+ age)’. What is the purpose and rationale behind?
A small paragraph regarding the aforementioned issue has been addressed in the revised manuscript at the empirical study section.
R2#9: Some statements are difficult to believe, for example: ‘They did not face any particular difficulties in using them’ (p. 11) referring to the first group (4-6 y.ol). How do the authors know that the small children did not face any difficulty?
A small paragraph regarding the aforementioned issue has been addressed in the revised manuscript at the empirical study section.
R2#10: The authors seem to adopt sometimes an overenthusiastic view of their results, which could be based on a biassed approach. For example ‘For the second group, we were delighted to find out that the students were enthusiastic about these apps.’ (p. 12) And also in the statement ‘Another essential and extremely gratifying finding is that students who were weak’ (p. 12)
The aforementioned changes have been addressed in the revised manuscript.
R2#11: I believe the authors should definitely include one specific section about context and research participants, method, instruments, etc before the results. There is a scarcity of details which are included under the section ‘empirical study’. But this section apparently combines method and context with research analysis, so the reader may be confused about some results as the data is not sequentially explained.
The aforementioned changes have been addressed in the revised manuscript.
R2#12: The research procedure is not clearly explained: ‘In the end, to complete the evaluation process, the students filled out a questionnaire’ (p. 14)
All participants completed the questionnaire? Small children (4-6 y.o.) too? How did they complete the questionnaire? Which one (use an Appendix)?
The aforementioned changes have been addressed in the revised manuscript.
R2#13: IMAGES. I believe some images may not be needed as they do not convey any relevant data about the research.
The aforementioned changes have been addressed in the revised manuscript.
R2#14: DATA ANALYSIS. There is no specific section about this, it is included in the empirical study but the data should not be only based on graphs (Google forms). They should also contain the reliability of the instrument (questionnaire) for each group, the SD, Means, etc.
The aforementioned changes have been addressed in the revised manuscript.
R2#15: And better explain some of these graphs such as why did the authors use a 10-point scale and what ‘type’ of technology does each graph refer to (Mondly, Drops), for example Fig. 15?
The aforementioned changes have been addressed in the revised manuscript.
R2#16: CONCLUSIONS. The authors should separate the data results from the conclusions. The result analysis should be related to previous works as there is no reference to previous literature after the introductory section. The conclusions are very general and not specifically related to the digital tools the authors allegedly used in their research.
The aforementioned changes have been addressed in the revised manuscript.
R2#17: The authors should state some research limitations.
Some minor English problems observed:
‘Students point to the letters thanks’ (p. 7) ‘thanks’
‘It’s a mandatory investment’ (p. 8) no contraction
‘The educational tools described above, enable students’ (p. 8) no comma
The aforementioned changes have been addressed in the revised manuscript.
We thank R2 for his response.

Round 2
Reviewer 1 Report
Dear authors,
thank you very much for your revision. In my opinion, this article was improved according to comments.
Reviewer 2 Report
Thanks for the revised version.